# Acute and Subchronic Toxicity Studies of Aristolochic Acid A in Tianfu Broilers

**DOI:** 10.3390/ani11061556

**Published:** 2021-05-27

**Authors:** Dan Xu, Chonglin Ran, Lizi Yin, Juchun Lin, Hualin Fu, Xi Peng, Xiaoling Zhao, Gang Shu

**Affiliations:** 1Farm Animal Genetic Resources Exploration and Innovation Key Laboratory of Sichuan Province, Sichuan Agricultural University, Chengdu 610000, China; 13086605395@163.com (D.X.); zhaoxiaoling@sicau.edu.cn (X.Z.); 2Department of Veterinary Medicine, Sichuan Agricultural University, Chengdu 610000, China; rcl960628@163.com (C.R.); yinlizi@hotmail.com (L.Y.); juchunlin@126.com (J.L.); fuhl.sicau@163.com (H.F.); pengxi197313@163.com (X.P.)

**Keywords:** Tianfu broiler, aristolochic acid, oxidative stress, kidney, subchronic toxicity

## Abstract

**Simple Summary:**

Aristolochic acid (AA) is a chemical compound present in traditional Chinese medicine, which is widely used for anti-infection, anti-viral and anti-bacterial treatment, as antibiotics were banned in the poultry industry. However, long-term use of this drug in high doses can cause harmful damage to the kidneys of animals. Therefore, in this study, the acute toxicity test and subchronic toxicity test of AA were carried out in order to explore the nephrotoxicity mechanism of AA on Tianfu broilers. In this study, the LD50 of AA to male Tianfu broilers was 14.52 mg/kg. Subchronic exposure to high doses of AA in broilers can cause nephrotoxicity by breaking the redox balance to form oxidative stress, along with promoting apoptosis of renal cells. In conclusion, AA has been found to damage broilers’ kidneys in high doses. This study provides suggestions for the clinical application of traditional Chinese medicine containing AA in the poultry industry.

**Abstract:**

Aristolochic acid (AA) is one of the components of some traditional Chinese medicines, which has high toxic potential in animals, leading to huge economic losses in the breeding industry. The purpose of this study is to evaluate the toxicology of AA on Tianfu broilers through acute and subchronic toxicity tests. The results showed that the median lethal dose of AA to Tianfu broilers was 14.52 mg/kg. After continuous intraperitoneal injection of AA solution (1.452 mg/kg) for 28 days, the swollen and necrotic renal tubular epithelial cells were histologically observed; in addition, blood urea nitrogen (BUN) and creatinine (Cre) were significantly increased, indicating AA could induce serious kidney lesions in broilers. Moreover, the ROS, the apoptosis rate and the depolarization rate of the mitochondrial membrane potential of broilers’ renal cells increased. The results of QRT-PCR showed that AA reduced the mRNA expressions of HO-1, NQO1, Raf-1 and Bcl-2, while the expressions of Bax and Caspase-3 increased, which show that AA aroused oxidative stress and promoted the apoptosis of renal cells. In conclusion, AA has been found to damage broilers’ kidneys by breaking the redox balance to form oxidative stress, along with promoting apoptosis of renal cells.

## 1. Introduction

Aristolochic acid A (AA, 8-methoxy-3,4-methylenedioxy-10-nitrophenanthrene-1-carboxylic acid, Figure 1) is a chemical compound present in plants of the Aristolochia, Bragantia and Asarum species. More than ten years ago, traditional Chinese medicines containing AA, such as houttuynia, were widely used in China. Houttuynia was often used for anti-infection, anti-viral and anti-bacterial treatment, as well as for the termination of pregnancy [1]. However, some scientists have pointed out that aristolochic acids in traditional Chinese medicines have a dangerous effect [2]. Using these drugs for a long time can cause harmful damage to animals’ kidneys [3]. The renal damage caused by AA is mostly subchronic or chronic through the pathways of oxidative stress injury and apoptosis via increasing the level of hydrogen peroxide in NRK-52E cells and up-regulating the mRNA expression of caspase-3 [4]. AA was also reported to arouse inflammatory reaction in the kidneys, which infiltrated a large number of inflammatory cells in the renal cortex and medulla, leading to a large area of renal interstitial fibrosis [5]. Other reports suggested that AA can quickly enter the renal tubular epithelial cells and accumulate without being discharged. Then, they react with residues to form DNA adducts of aristolochic acid lactam. Moreover, the DNA adducts cause TP53 heterotopic mutation from A:T to T:A, which is difficult to recognize in the whole genome nucleotides and difficult to remove for repairs, leading to an increased rate of mutation [6]. In addition, some literature confirmed that aristolochic acid A is the main toxic component of the aristolochic acids [7].

In China, the poultry industry has developed rapidly in recent years, but intensive farming has brought some disadvantages, leading to the high incidence of some diseases. At present, due to the prohibition of antibiotics, traditional Chinese medicine has been widely used as an alternative to allopathic drugs. Many traditional Chinese medicines containing AA have been used as alternatives to antibiotic drugs to treat poultry diseases. However, due to its nephrotoxicity associated with interstitial fibrosis, hyperproteinemia, severe anemia, uremia and carcinoma, AA and AA-containing herbs may potentially harm the development of the poultry industry [8]. At present, there is no study on the toxicity of AA in broilers. Therefore, it is necessary to determine the toxicity of AA and establish criteria for selecting a safe dose.

In this study, the acute toxicity test and subchronic toxicity test of AA were carried out on Tianfu broilers. The acute toxicity of AA to broilers was tested via the modified Coriolis method, by calculating the half lethal dose (LD50). The subchronic toxicity test was carried out according to LD50 to explore the nephrotoxicity mechanism of AA damage to Tianfu broilers, and to provide suggestions for the clinical use of Chinese medicine containing AA in the broiler breeding industry.

## 2. Material and Methods

### 2.1. Experimental Materials

Aristolochic acid A (AA, HPLC > 98%) was purchased from Chengdu Rifens Technology Co., Ltd., Sichuan, China. The chicken-specific ELISA assay kits for blood urea nitrogen (BUN), creatinine (Cre), malondialdehyde (MDA), superoxidase dismutase (SOD) and reduced glutathione (GSH) were purchased from Nanjing Jiancheng Bioengineering Institute (Nanjing, China).

### 2.2. Experimental Animals

The animal experiment was performed at the poultry farm of Sichuan Agricultural University. All the procedures were conducted in accordance with the national standard Laboratory Animal Requirements of Environment and Housing Facilities (GB 14925–2001), and animal handling and care were approved by the Sichuan Agricultural University’s Institutional Animal Care and Use Committee under permit number DYY-2018203007. A total of 96, 1-day-old male Tianfu broilers, 38.83 ± 3.45 g of initial BW, were used in this study (Poultry Breeding Farm of Sichuan Agricultural University, Chengdu, China). Experimental chickens were raised in cages in a well-ventilated house, and the relative humidity was maintained at 50%. The photoperiod was maintained at 24 h for the first 14 days and later reduced to 20 h for the subsequent days. The temperature was 34 °C initially and was subsequently decreased by 2 °C per week to 24 °C, which was maintained until the end of the experiment. The birds had free access to feed and water throughout the entire experimental period. The formulation and nutrient levels of their regular diets were based on the National Research Council requirements for chickens (Appendix A) [9].

### 2.3. Acute Poisoning with Aristolochic Acid A in Tianfu Broilers

A total of 56, 1-day-old male Tianfu broilers were randomly divided into 7 groups. The doses for the groups were 54.08 mg/kg (group 1), 34.89 mg/kg (group 2), 22.51 mg/kg (group 3), 14.52 mg/kg (group 4), 9.37 mg/kg (group 5), 6.05 mg/kg (group 6) and 3.90 mg/kg (group 7). The mental state of each group was observed for 14 days just after the intraperitoneal injection, and the mortality rate was also recorded. An aristolochic acid A solution (pH = 7.8) was prepared with ultrapure water. The LD50 of aristolochic acid A in Tianfu broilers was calculated according to the mortality of each group and the modified Coriolis method. The formula was as follows (1):logLD50 = XK − i(Σ*^P^* − 0.5)(1)

XK: Maximum dose group dose pair value, i: Difference of dose pair values between two adjacent groups, Σ*^P^*: total mortality in each group [10].

### 2.4. Subchronic Poisoning with Aristolochic Acid A in Tianfu Broilers

To further characterize the toxicological potential of AA, as well as to evaluate the toxic mechanisms on the animal biochemical profile and renal histopathology, we selected 1/100 LD50, 1/50 LD50 and 1/10 LD50 as the experimental doses and carried out a 28-day subchronic toxicity experiment. 

#### 2.4.1. Grouping and Treatment of Experimental Animals

According to the results of the acute toxicity experiment, we chose 1/100 LD50, 1/50 LD50 and 1/10 LD50 as the doses of the subchronic toxicity experiment. A total of 40, 1-day-old male broilers were randomly divided into four groups. Broilers in the control group (CG), low-dose group (LAG), middle-dose group (MAG) and high-dose group (HAG) were injected intraperitoneally with normal saline, aristolochic acid A solution (1/100 LD50), aristolochic acid A solution (1/50 LD50) and aristolochic acid A solution (1/10 LD50) every 24 h, respectively. The experimental period lasted for 28 days, and on the 29th day, the broilers were sacrificed and blood samples were collected for biochemical analysis. Kidney samples were weighed and flushed with 0.9% NaCl and later prepared for further analysis, and the renal index was expressed as the ratio of organ weight to body weight (g/kg).

#### 2.4.2. Biochemical Analysis

The levels of Cre and BUN in serum, as well as the contents of MDA, GSH and SOD in the kidneys, were detected using specific ELISA kits.

#### 2.4.3. Histopathology

Histological analysis was performed adhering to the guidelines described in the previous study [11]. The kidney samples were fixed in 4% (*wt/vol*) buffered paraformaldehyde for 24 h. The trimmed samples were dehydrated, cleared and then embedded in paraffin. The samples were sectioned into 5 μm slices by using a RM2235 microtome (Leica, Munich, Germany), flattened onto glass slides and then dried. After dewaxing with xylene, the sections were stained with hematoxylin and eosin (Thermo, Waltham, MA, USA) and then sealed with neutral resin. The histopathological change of the kidneys was visualized under a CX22 microscope (Olympus, Tokyo Metropolitan, Japan), and a DM1000 microimaging system (Leica, Munich, Germany) was used to capture images.

#### 2.4.4. Reactive Oxygen Species (ROS), Mitochondrial Membrane Potential (MMP) and Apoptosis of Kidney by Flow Cytometry

Kidney samples were cut, centrifuged and suspended to 1 × 10^6^ cell/L in frozen PBS solution for the following flow cytometry test. The prepared ileal cells were incubated for 15 min at 37 °C with DCFH-DA (total ROS assay kit from Nanjing Jiancheng Bioengineering Institute (Nanjing, China)) in the dark. The reactive oxygen species (ROS, %) generation in the ileum was determined using a flow cytometer. Another sample was incubated for 15 min at 37 °C with JC-1 in the dark, and the MMP was measured using flow cytometry analysis. The result of MMP was described as mitochondrial depolarization ratio. Furthermore, the cells were incubated for 15 min at 37 °C with 5 μL Annexin V-FITC and 5 μL PI. The percentage of apoptosis was measured using flow cytometry (Bio-Rad, Hercules, CA, USA), and all the data collected were analyzed by Kaluza 2.1 Software, Beckman Coulter, CA, USA.

#### 2.4.5. Fluorescence Real-Time Quantification PCR (QRT-PCR)

The kidney samples were washed with pre-frozen DEPC water and immediately frozen in liquid nitrogen and stored at −80 °C. Approximately 60 mg of preserved sample was ground thoroughly with liquid nitrogen in a precooled mortar. Total RNA was extracted with TRIzol (Invitrogen, Shanghai, China). The RNA concentration was determined by a nucleic acid protein analyzer, of which the D260/D280 range eligible for reverse transcription was 1.8–2.0. The cDNA was stored at −80 °C.

According to the specific steps of the SYBR Green Remix Ex TaqTM kit specification of TaKaRa, QRT-PCR was used to detect the expression levels of the following genes: Claudin1, NQO1, HO-1, Caspase 3, Bax, Bcl-2 and Raf-1. β-actin was used as the endogenous control to normalize the expression of genes. We chose the CG group (blank group) as a reference group (value = 1), and the fold change of gene expressions was quantified using the 2^−ΔΔCt^ method, where ΔCt = Ct _target gene_ − Ct _housekeeping gene_, and ΔΔCt = ΔCt − ΔCt _reference_. All primers (Appendix A) were designed using Premier 5 (PREMIER Biosoft International, New York, NY, USA) and synthesized by Chengke BioTech Co., Ltd, Guangzhou, China.

### 2.5. Statistical Analyses

Data were analyzed via one-way analysis of variance (ANOVA) with statistical software SPSS 19.0 (SPSS Inc., Chicago, IL, USA). Duncan’s multiple range test was used to determine the differences among treatment groups. All parameters determined in this study were presented as mean ± standard error (mean ± SE). Significance was determined at a level *p* < 0.05.

## 3. Results

### 3.1. Acute Poisoning with Aristolochic Acid A in Tianfu Broilers

After intraperitoneal injection, each group had different clinical symptoms. In group 1, the broilers showed severe clinical symptoms, such as lethargy, shortness of breath, curling and decreased feed and water intake. The same symptoms were also recorded in group 2. Mortalities in groups 1 and 2 were observed at 8 h and 11 h after the injection, respectively. Meanwhile, clinical symptoms observed in groups 3, 4, 5 and 6 were similar with a weak remission, as compared to group 1; notably, the broilers in group 7 recorded no death and were without any clinical symptoms. As shown in Table 1, during the 14 days of observation, the broilers in group 1 (*n* = 8) were all dead, and 4 mortalities were observed in groups 3, 4 and 5. Two mortalities were recorded in group 6, but no mortalities were observed in group 7 (Table 1).

The LD50 of aristolochic acid A in male Tianfu broilers was calculated according to the modified Coriolis method and the mortality of broilers. LD50 = 14.52 mg/kg, LD50 is within the confidence interval.

### 3.2. Subchronic Poisoning with Aristolochic Acid A in Tianfu Broilers

#### 3.2.1. Aristolochic Acid A Increased Renal Index in Broilers

As shown in Figure 2, after 28 days of intraperitoneal injection of aristolochic acid A, the kidneys in the HAG became swollen, tense and fragile. As listed in Figure 3, the renal indexes in both the MAG and HAG were significantly higher than that in the CG (*p* < 0.05).

#### 3.2.2. Aristolochic Acid A Changed Renal Function in Broilers

As shown in Figure 4, renal function was accessed by measuring the content of Cre and BUN in serum. The levels of Cre in the LAG, MAG and HAG were significantly increased compared with the control group (*p* < 0.05), while the HAG recorded the highest value (*p* < 0.05). Compared with the CG, the BUN level in the high-dose group was significantly increased (*p* < 0.05).

#### 3.2.3. Aristolochic Acid A Destroyed Normal Renal Tissue Structure

According to the results of the histopathological examination, there was no obvious histopathological damage in the CG (Figure 5A). In the three drug treatment groups, the toxic pathological injuries were characterized as glomerular hyperemia, interstitial congestion between renal tubules and degeneration and necrosis of renal tubular epithelia cells, all of which showed a dose–effect relationship. In the LAG, several renal tubular epithelial cells in the medullary area had hydropic degeneration (Figure 5B). In the MAG, the volume of glomeruli increased significantly due to hyperemia, and the renal vesicles became narrow; the capillaries in the tubulointerstitium were dilated and congested; extensive hydropic degeneration of renal tubular epithelial cells was observed, and several cells became enlarged with much denser nuclei and increased eosinophilic small powdery grains in the cytoplasm, which often were separated from the adjacent normal cells (Figure 5C). The histopathological lesions were more severe in the HAG than in those of the MAG. Firstly, because of severe congestion, it was observed that the glomeruli were obviously enlarged, the renal vesicles were almost atresia and the interstitial venules and capillaries were significantly expanded and filled with red blood cells. Secondly, even though there were increased necrobiotic cells with dense nuclei and eosinophilic cytoplasm, some cells were necrotic and shed from the basement tubular membrane (Figure 5D).

#### 3.2.4. Aristolochic Acid A Destroyed the Antioxidant System to Arouse Oxidative Stress

Figure 6 illustrates that the level of SOD in the HAG was significantly lower than that in the CG (*p* < 0.05), and the levels of GSH in the LAG, MAG and HAG were significantly lower than that in the CG (*p* < 0.05). Compared with the CG, the levels of MDA and ROS in both the MAG and HAG increased significantly (*p* < 0.05).

#### 3.2.5. Aristolochic Acid A Promotes Renal Apoptosis and Oxidative Stress in Broilers

Figure 7 shows that the apoptotic rate of renal cells in both the MAG and HAG increased significantly (*p* < 0.05). Compared with CG, the mitochondrial depolarization ratios of the LAG, MAG and HAG increased significantly (*p* < 0.05).

According to Figure 8, after 28 days of AA treatment, compared with the CG, the expressions of Bax and Caspase 3 in the HAG were significantly higher (*p* < 0.05), while the expressions of Bcl-2, HO-1, Raf-1, and NQO1 were significantly lower (*p* < 0.05).

## 4. Discussion

AA is a chemical compound present in many traditional Chinese medicines with a high toxic potential to animals, causing acute tubular necrosis. Cosyns et al. reported that chronic administration of AA induced renal interstitial fibrosis in rabbits [12]. Mengs, U. et al. pointed out that high doses of AA appeared carcinogenic and nephrotoxic in varied species (including rat, mouse, rabbit, pig and human), and may promote renal and extrarenal fibrosis in humans [13,14]. Consistent with the previous studies, our study found that AA can induce oxidative stress injury in the kidneys of Tianfu broilers, leading to apoptosis of renal cells and, eventually, kidney injury.

Mortality is a clear sign of toxicity, but other variables may indicate more subtle adverse effects, such as clinical signs of toxicity [15]. Mengs, U. pointed out that high-dose intravenous administration of AA caused death in rats within 15 days, and the kidneys of rats became swollen with the predominant feature of severe renal tubule necrosis [16]. Similar clinical symptoms were observed in broilers in this study. After counting the number of deaths of broilers in different dose groups, LD50 of AA in male Tianfu broilers was calculated according to the modified Coriolis method, LD50 = 14.52 mg/kg.

The results of the subchronic toxicity experiment revealed the toxicological potential of AA, as well as the toxic mechanisms on broilers’ biochemical profiles and renal histopathology. It has been reported that after treatment with aristolochic acid I in mice, the kidneys showed significant pathological changes, including swelling, darkness, tightness and fragility [17]. The same renal changes in the present study suggest that AA could induce nephrotoxicity in broilers as well. Moreover, since the alteration of organ relative weight is also considered an evidence of toxicity, the increase in renal index in the AA group may be related to the swelling of the kidneys [18]. BUN and Cre are important indexes of renal function [19]. AA can increase the release of BUN and Cre in the serum of mice to induce renal dysfunction, which is consistent with our results [17]. The present study showed that AA can damage broilers’ kidneys by increasing BUN and Cre levels. Furthermore, histological analysis also indicated that AA can cause severe renal injury with obvious necrosis of renal tubular epithelial cells, which is in accordance with the report of Pozdzik et al. [5].

Emerging evidence has demonstrated that AA induces renal injury by increasing the level of hydrogen peroxide and causing oxidative stress injury [4]. Oxidation and reduction is a procedure for energy production, which is common to many fundamental responses of organisms. Oxidative stress disrupts the redox balance (reduction–oxidation) with high levels of ROS generation [20]. In our current study, increased levels of renal ROS and MDA, along with decreased activity of renal antioxidant compounds (SOD and GSH), were observed in the AA treatment groups, which revealed that AA may break the redox balance of broilers. This was consistent with previous findings [21]. AA-induced oxidative stress was also evidenced by the downregulation of HO-1 and NQO1. HO-1 exerts a protective role in many diseases through reducing ROS accumulation and inhibiting mitochondrial structural dysfunction [22], while NQO1, as an antioxidant gene, plays an essential function in resisting oxidative injury [23]. Specifically, the results showed a marked decrease in the expressions of HO-1 and NQO1 genes following AA treatment, suggesting that AA could break the redox balance in vivo to cause oxidative damage in broiler kidneys [24]. Consistently, Matsui et al. indicated that AA induces nephrotoxicity through oxidative stress pathways in mice [25]. Moreover, AA can induce mitochondrial dysfunction by significantly increasing the production of mitochondrial free radicals with the increase in the active oxygen species to damage the integrity and morphology of mitochondria, as well as to decrease the electron density of the matrix in renal cells, and eventually lead to cell death [26]. Our study found that continuous exposure to high doses of AA increased the depolarization rate of broiler renal cells. Therefore, this further revealed that AA induced renal cytotoxicity by destroying the integrity of renal mitochondria and increasing ROS formation.

Another certain cytotoxicity induced by oxidative stress is apoptosis. Excessive ROS destroys the normal mitochondrial structure and function by increasing the opening of mitochondrial membrane permeability transition pores [27]. In our study, AA elevated the incidence of apoptosis in broiler kidneys, which is in line with the work done by Hsin, who reported that AA can cause a rapid increase in apoptosis rate by arousing the endoplasmic reticulum and mitochondrial stress, as well as caspases activation [28]. The mRNA expression of the four apoptosis-related biomarkers (Bcl-2, Bax, Raf-1 and caspase 3) also supported this result since AA had reduced the levels of Raf-1 and Bcl-2, as well as increased the levels of caspase 3 and Bax to arouse apoptosis [29]. High expression of Bcl-2 can protect mitochondria by maintaining Ca^2+^ in a stable state to inhibit apoptosis [30]. Bax, which plays an antagonistic role with Bcl-2, destroys the integrity of the mitochondrial membrane to accelerate apoptosis. Raf-1 is the downstream gene of Bcl-2, which exerts an anti-apoptosis role by transmitting the signal of apoptosis inhibition to mitochondria [31,32]. Caspase-3, as an executor of apoptosis (which can be regulated by Bax and Bcl-2 downstream), changes the mitochondrial membrane permeability to arouse apoptosis [33]. Up-regulation of caspase 3 and Bax, along with down-regulation of Bcl-2 and Raf-1 in the AA treatment groups in our study, further indicated that AA triggered renal apoptosis through the mitochondria-mediated apoptosis pathway. However, further studies are required to investigate the specific toxicology mechanisms of AA, especially the inflammation pathway.

## 5. Conclusions

This study demonstrated medium AA toxicity after acute and subchronic oral exposure to the broilers. The LD50 of AA to the male Tianfu broilers was established as 14.52 mg/kg. AA elicited renal toxicity after subchronic exposure in the broilers, as it can break the redox balance of the kidneys, which further damaged the integrity of the mitochondria and aroused apoptosis, all of which eventually led to kidney damage (as summarized in Figure 9). This study has brought information about the clinical dosage of Chinese medicine containing AA in broilers.

## Figures and Tables

**Figure 1 animals-11-01556-f001:**
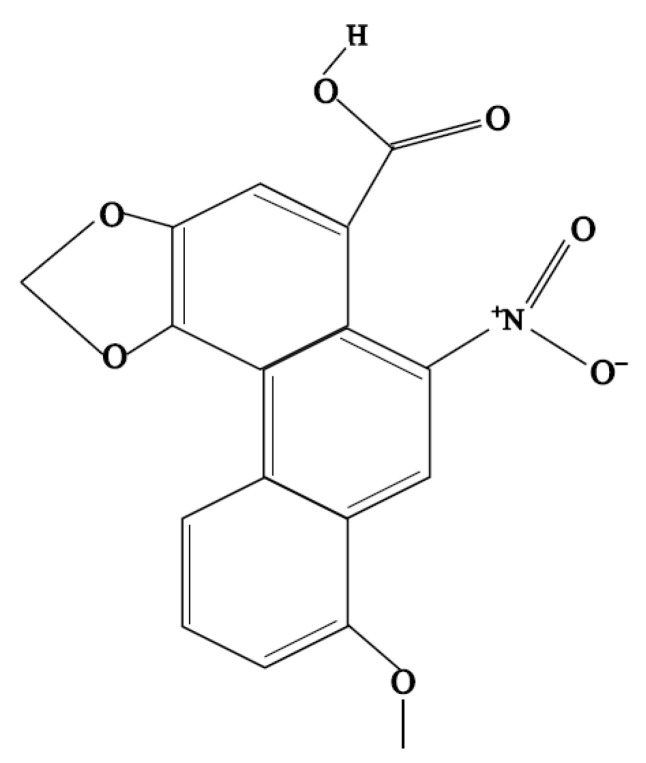
Chemical structural of aristolochic acid A.

**Figure 2 animals-11-01556-f002:**
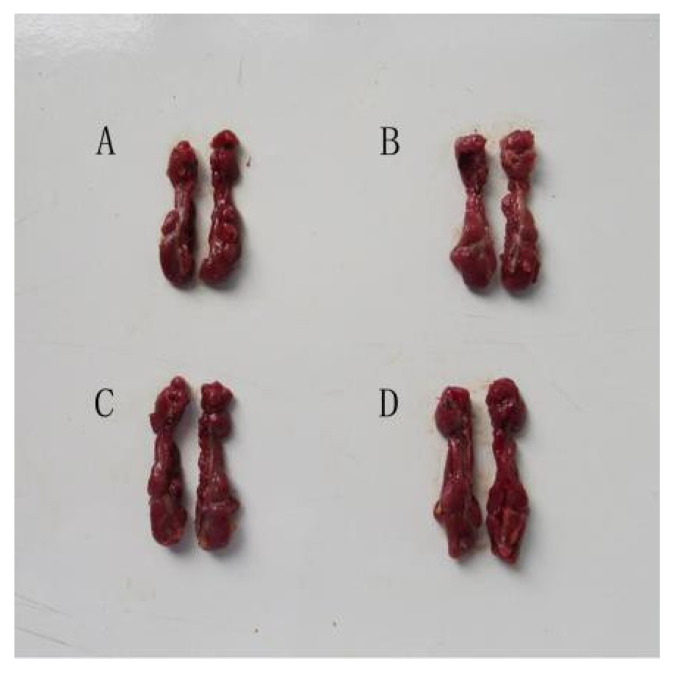
Renal morphology in different groups. (**A**) CG group; (**B**) LAG group; (**C**) MAG group; (**D**) HAG group.

**Figure 3 animals-11-01556-f003:**
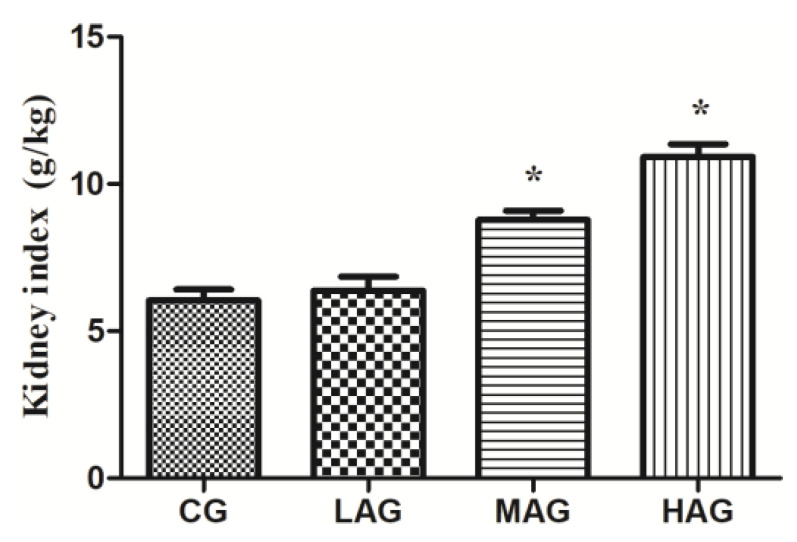
Kidney index of broilers in different groups. * *p* < 0.05, compared with the control group (CG).

**Figure 4 animals-11-01556-f004:**
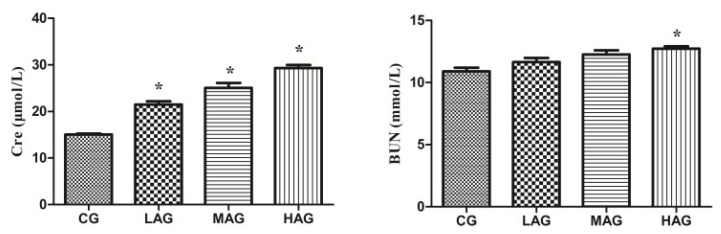
Effects of aristolochic acid A on renal function of Tianfu broilers. * *p* < 0.05, compared with the control group (CG). BUN: blood urea nitrogen (μmol/L), Cre: creatine (mmol/L).

**Figure 5 animals-11-01556-f005:**
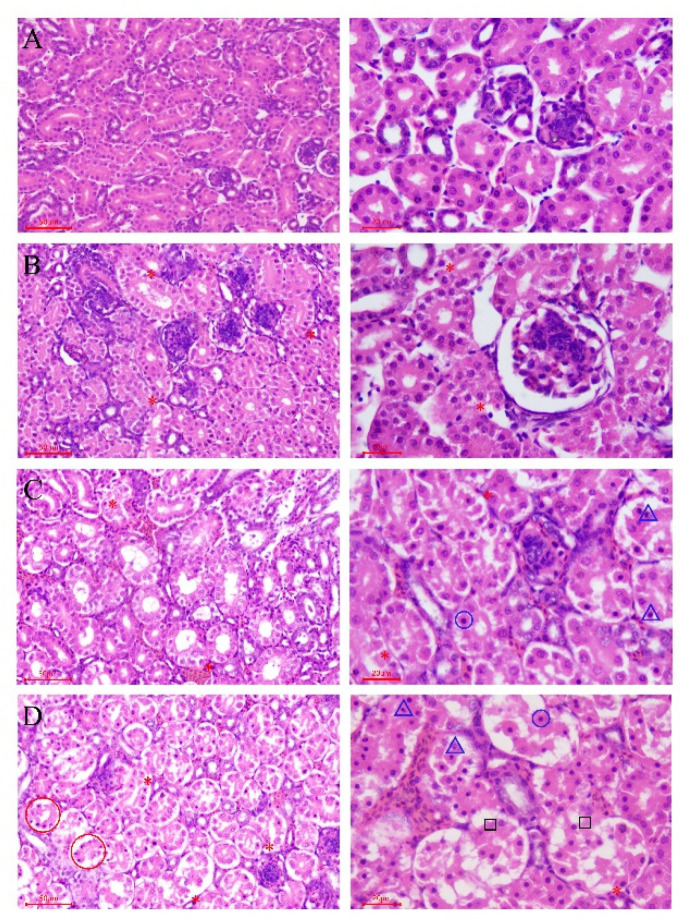
Pathological section of the kidney in different groups. (**A**) CG group; (**B**) LAG group; (**C**) MAG group; (**D**) HAG group. * Cell swelling. Blue circle: karyopyknosis. Blue triangle: karyorrhexis. Black quadrilateral: karyolysis. Red circle: all epithelial cells in a renal tubule were necrotic and shed from the basal layer. H.E. stain, on the left, magnification 200×; the right, magnification 400×.

**Figure 6 animals-11-01556-f006:**
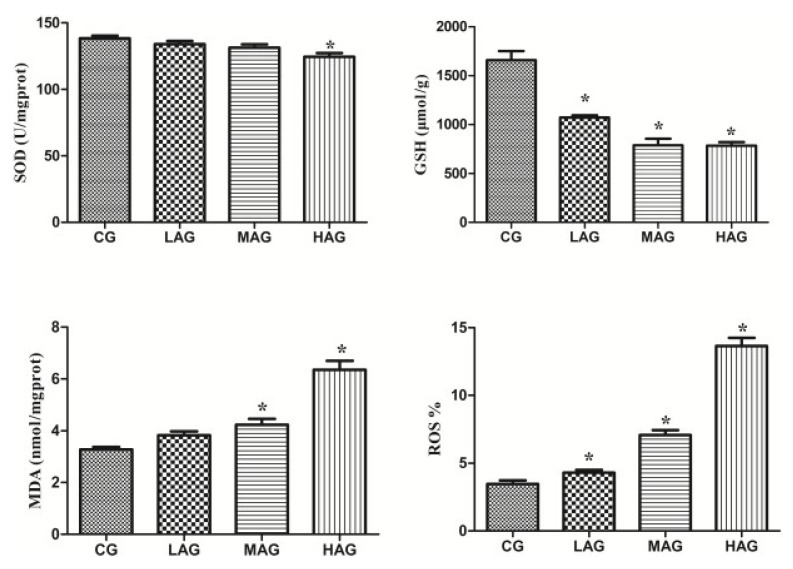
Effects of aristolochic acid A on the antioxidant system of Tianfu broilers. * *p* < 0.05, compared with the control group. (CG). MDA: malondialdehyde (mmol/L), SOD: superoxidase dismutase (U/mL), GSH: glutathione (μmol/g) and ROS: Reactive oxygen species (%).

**Figure 7 animals-11-01556-f007:**
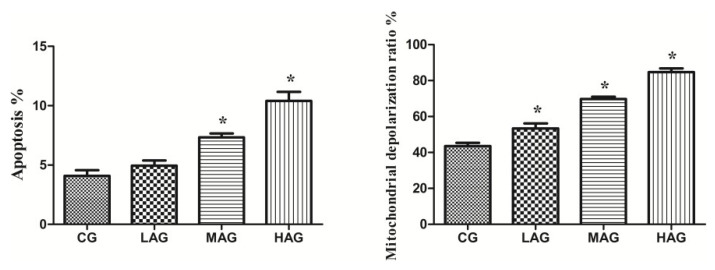
Comparison of mitochondrial membrane potential and apoptosis rate of renal cells in each group. * *p* < 0.05, compared with the control group (CG).

**Figure 8 animals-11-01556-f008:**
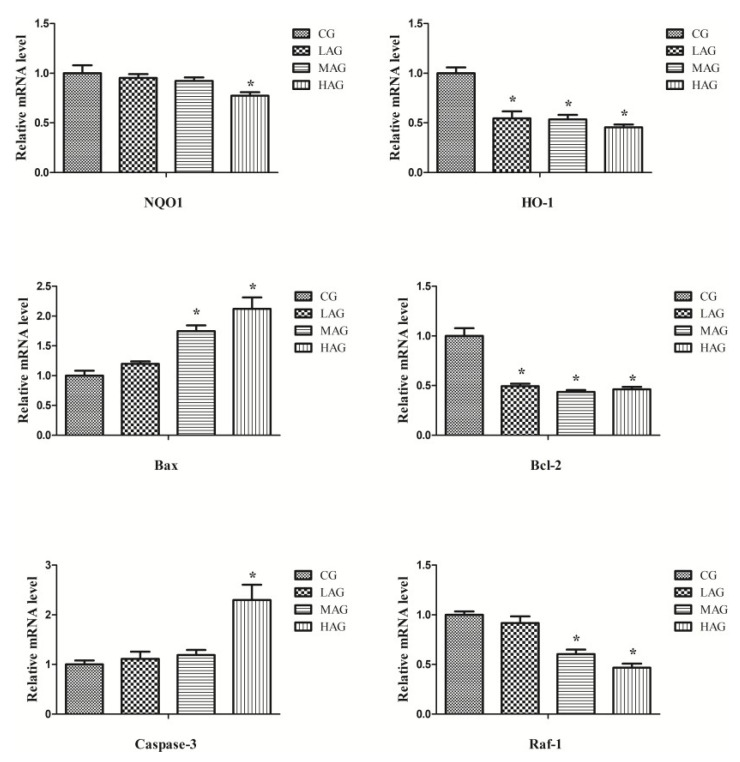
The mRNA expression of genes (NQO1, HO-1, Raf-1, Caspase 3, Bax and Bcl-2) of renal cells in each group. * *p* < 0.05, compared with the control group (CG).

**Figure 9 animals-11-01556-f009:**
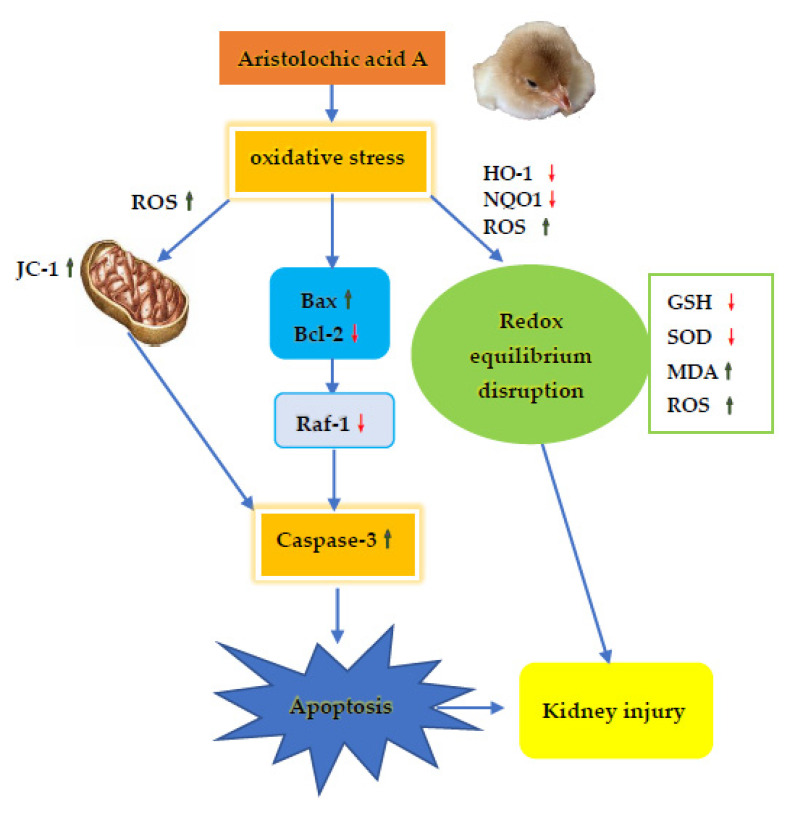
Schematic diagram of the possible mechanism of AA-induced renal injury in Tianfu broilers. The toxicological mechanism of AA-induced renal injury contains excessive apoptosis and oxidative stress damage.

**Table 1 animals-11-01556-t001:** Number and mortality of broilers after 14 days of administration (*n* = 8).

Group Number	1	2	3	4	5	6	7
Death	8	6	4	4	4	2	0
Mortality (%)	100	75	50	50	50	25	0

## Data Availability

The data presented in this study are available on request from the corresponding authors. The data are not publicly available due to privacy protection.

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
