# Peer review of "Acute and Subchronic Toxicity Studies of Aristolochic Acid A in Tianfu Broilers"

_animals, 2021, doi:10.3390/ani11061556_

Round 1
Reviewer 1 Report
Xu et a present an original research article entitled "Acute and subchronic toxicity studies of the Aristolochic acid A in Tianfu broilers".
They look at the effect of aristolochic acids (AA) , which has high toxic potential in animals, so an impact on cattle breeding. Using a model of male Tianfu broilers, they conclude that AA damages broilers’ kidneys by breaking the redox balance to form oxidative stress, along with promoting apoptosis of renal cells. The LD50 of AA to the male Tianfu broilers was established as14.52mg/kg.
Query: justify the sex of poultry used.
in RT-qPCR, when using 2-△△CT method, precise what was compared to what?
In the discuisson the others should discuss what is known about toxicity of AA in other ania pecies and human.
a final recapitulative figure would be useful to the reader.
Minor
What is the pont of 'Bars with different letters are statistically different (P < 0.05)' why use different letters?
line 171 more severe, not severer
Reviewer 2 Report
The article concerns the determination of aristolochic acid toxicity in broilers. The study was properly planned, the introduction is comprehensive, and the materials and methods, results and discussion are properly described. The conclusions are also adequate to the research performed. However, the very idea of the study is unconvincing to me, because aristolochic acid is considered toxic in humans, its harmful effect on the kidneys and liver, and its carcinogenic effect have been confirmed. Hence, I believe that drugs containing this compound should not be considered for use in farm animals as a treatment for diseases.
Some additional comments are presented below:
-
In the figures, the way to show statistically significant differences should be improved. It is difficult to understand what different letters above the bars mean.
-
All abbreviations used in the figures (especially under bars) should be explained in the description of the figures, in addition to the figure titles.
Reviewer 3 Report
The manuscript described toxicity studies of aristolochic acid (AA). This article is the first one to refer to the toxicity of AA in broilers. The authors revealed that AA induced oxidative stress injury in kidneys of broilers. Thus, these findings will be useful for zoology. Therefore, the manuscript is not too excellent to be published. In other words, the manuscript is so excellent that it should be published.
Comments
(1) AA reduced the mRNA expression of Bcl-2 and increased the expression of Bax and Caspase-3. Thus, it was suggested that AA promoted the apoptosis. How did AA promote the apoptosis with respect to signal transduction?
(2) AA is one of the components of some traditional Chinese medicines. However, AA showed high toxicities, particularly kidney toxicity, in animal. Was this resulted from species differences between human and animals?
(3) AA is likely to be relevant to human kidney toxicity. Is this the same toxic mechanism as the case in broilers?
(4) Please show the structure of AA.
(5) In lines of 35-36, “injection respectively” is preferable to be “injection, respectively”.
That is all.
Round 2
Reviewer 1 Report
changes ok